# Alimentary Risk of Mycotoxins for Humans and Animals

**DOI:** 10.3390/toxins13110822

**Published:** 2021-11-21

**Authors:** Jagoda Kępińska-Pacelik, Wioletta Biel

**Affiliations:** Department of Monogastric Animal Sciences, Division of Animal Nutrition and Food, West Pomeranian University of Technology in Szczecin, Klemensa Janickiego 29, 71-270 Szczecin, Poland; jagoda.kepinska-pacelik@zut.edu.pl

**Keywords:** feed, food, pet food, microbiological hazards, safety, mycotoxins

## Abstract

Mycotoxins can be found in many foods consumed by humans and animals. These substances are secondary metabolites of some fungi species and are resistant to technological processes (cooking, frying, baking, distillation, fermentation). They most often contaminate products of animal (beef, pork, poultry, lamb, fish, game meat, milk) and plant origin (cereals, processed cereals, vegetables, nuts). It is estimated that about 25% of the world’s harvest may be contaminated with mycotoxins. These substances damage crops and may cause mycotoxicosis. Many mycotoxins can be present in food, together with mold fungi, increasing the exposure of humans and animals to them. In this review we characterized the health risks caused by mycotoxins found in food, pet food and feed. The most important groups of mycotoxins are presented in terms of their toxicity and occurrence.

## 1. Introduction

Currently, despite many available methods of monitoring and preventing food contamination, there are threats that we are not able to completely eliminate. One of the sources of such threats are secondary metabolites of mold fungi—mycotoxins. The name is derived from the Greek word *mycos* (fungi) and Latin *toxicum* (poison). Many mycotoxins have been characterized, but the most important in terms of food safety are: aflatoxins, deoxynivalenol, fumonisins, zearalenone, and ochratoxin A. These basic forms of mycotoxins can be converted into metabolites with altered chemical structures and different physicochemical, chemical, and biological properties. Modified forms may arise in raw materials intended for food production and during technological processes [1]. Modified mycotoxins can also be produced by fungi or as part of an infected plant’s defense mechanism [2].

The interest in mycotoxins results mainly from their physicochemical properties, such as their high stability in changing environmental conditions and their high toxicity [3]. Even low levels of mycotoxins in food can lead to serious health consequences, especially because, as low-molecular and thermostable substances, they are resistant to most technological processes, e.g., cooking, frying, baking, distillation, and fermentation [4]. Every day, both humans and animals are exposed to low-doses of mycotoxins [5,6,7].

The occurrence of mycotoxins not only affects the health of consumers, but also exerts an impact on global trade. Long-term consumption of mycotoxin-containing pet-food and feed by animals may decrease the production of milk, meat, or wool, and alterations to the reproduction and growth of animals can be detected. The potential toxicity of mycotoxins on the biological performance of animals, such as pigs and chickens, has been reported in the literature. Additionally, weight loss, immunosuppression with increased prevalence of diseases, and decreased reproductive capacity are some of the main risks associated with the consumption of foods contaminated by secondary metabolites of fungi [8].

Therefore, the aim of this review is to highlight the health risks posed by mycotoxins in food, pet food, and feed. The most important groups of mycotoxins in terms of their toxicity and occurrence are presented.

## 2. Mycotoxin Contamination

It is estimated that 25% of world plant production and 20% in the European Union may be contaminated with mycotoxins [9]. They are most often produced by fungi of the genera *Aspergillus*, *Penicillium*, *Fusarium*, and *Alternaria* (Figure 1). Their synthesis depends largely on the properties of fungal strains (physiological, genetic, and biochemical), as well as external factors, such as humidity and temperature. This means that the level of mold contamination depends, inter alia, on local weather conditions, because high humidity, optimal temperatures, and specific environments affect their formation.

Most often, mycotoxins contaminate plant-based products, such as cereals, vegetables, dried fruits, nuts, coffee, cocoa, and tea. They can even be present in wine and beer [13]. However, plant products are not the only sources of mycotoxins. This can also apply to raw materials and products of animal origin (e.g., milk, meat) [14,15,16]. Mold toxins are characterized by a low molecular weight (M < 1.5 kDa). They can be absorbed through the digestive and respiratory tracts, as well as through the skin and mucous membranes. Individual mycotoxins can be genotoxic, mutagenic, and teratogenic. Diseases caused by mycotoxins are called mycotoxicosis [17]. In the European Union, the issue of the maximum levels of certain contaminants in foodstuffs is regulated by the Commission Regulation (EC) No. 1881/2006 [18]. The acceptable levels of secondary metabolites of mold fungi in selected foodstuffs are presented in Table 1.

As mentioned earlier, mycotoxins contaminate many plant-based products. An example is green tea leaves, the infusion of which is one of the most consumed beverages in the world. Research [19] found that of the green tea samples available in Morocco, 56% was contaminated with at least one mycotoxin. The highest level was found for ZEN. Although many mycotoxins coexisted in the samples, probable daily intake estimates showed that the ingestion of mycotoxins through consumption of green tea does not pose a risk to the population.

## 3. Harmful Effect of Mycotoxins for Humans and Animals

### 3.1. Classification

The toxic effect of metabolites of mold fungi is diverse and depends on the chemical structure [20]. Depending on the damaged organ, mycotoxins can be divided into:
‒Dermatotoxins—damaging mucous membranes and skin;‒Hepatotoxins—leading to liver damage;‒Cardiotoxins—causing cardiovascular diseases;‒Nephrotoxins—damaging the kidneys;‒Neurotoxins—affecting the central nervous system;‒Pulmotoxins—causing pulmonary edema.

In addition, there are also a different classification of mycotoxins:
‒Immunotoxins—negatively affecting the immunological system;‒Micohormones—negatively influencing hormonal regulation;‒Carcinogenic compounds—leading to the formation of neoplasms.

### 3.2. Aflatoxins

Different classes of aflatoxins (AF), such as B_1_, B_2_, G_1_, and G_2_ can be recognized. The capital letters refer to the color they take under UV light (blue—B, green—G). AF can be found mainly in peanuts, maize seeds, and oilseeds. AFB_1_ is the most prevalent and most toxic compound among AFs. In the liver, it is transformed into AFM_1_, which is then excreted into the milk of lactating mammals, including dairy animals. AFM_1_ has been shown to be cause of both acute and chronic toxicosis. The presence of AFM_1_ in milk and dairy products represents a worldwide concern since even small amounts of this metabolite may be of importance as long-term exposure is concerned [21].

AF can also be found in meat. The plague of 100,000 turkeys in 1960 on poultry farms in England resulted in a fundamental revision of the assessment of the effects of fungal metabolites on other organisms and increased interest in them, as it turned out that disease “X” was caused by AF. AFs are considered to be the best-studied mycotoxins [22]. The consequences of AF poisoning vary, mainly due to the amount of the dose and whether it is short-term or chronic contact. The effects of poisoning are serious and can be fatal. The most common symptoms are abdominal pain, vomiting, hemorrhage, liver damage, growth and development impairment, immunodeficiency, pulmonary and limb edema, weakening of blood vessels, and coma [22,23,24,25,26]. Among the mycotoxins known so far, AF has the strongest carcinogenic effect. Its chronic presence in food products supplied to the body contributes to the formation of liver cancer [20]. The International Agency for Research on Cancer (IARC) has classified AFB_1_ as “group I” due to its high toxicity, teratogenicity, hepatocarcinogenicity, and mutagenicity. AFB_1_ is converted into its fundamental hydroxylated metabolite called AFM_1_ in the liver of livestock by a superfamily of enzymes called cytochrome P450 and ingestion of feed contaminated with AFB_1_ can cause excretion of aflatoxin M_1_ in milk. Depending on toxicity and carcinogenicity, in 2012, IARC has also classified AFM_1_ as “group I” [27,28].

The great toxicity of AFB_1_ is based on its bioactivation process. AFB_1_ is biologically activated by a number of P450 cytochromes (CYPs), namely CYP1A and CYP3A, to an extremely reactive and electrophilic derivative, AFB1-8,9-epoxide (AFBO), which binds DNA and proteins. Such AFBO-DNA and AFBO-protein adducts inhibit RNA and protein synthesis, ultimately leading to severe toxicity and eventually cancer development. The animal intake of AFs always accompanies the production of reactive oxygen species (ROS) and the resulting oxidative damage may be a major trigger of detrimental outcomes [29].

The effects of AFB_1_ on animals vary according to the concentration and duration of contact with mycotoxin in food. High concentrations of this toxin are lethal, moderate concentrations lead to chronic poisoning. Since about one-fifteenth of the consumed AFB_1_ is excreted in milk as AFM_1_, and different heat treatments used in the preparation of different dairy products do not reduce the amount of AFM_1_, there is always the possibility of poisoning with this toxin by consuming contaminated milk [30].

### 3.3. Ochratoxin A

Exposure to ochratoxins comes from the consumption of foodstuffs of plant origin (grape juice, wine, coffee, spices, dried fruits, chestnuts, cereal-based products, e.g., whole-grain breads), and animal origin, e.g., pork blood-based products [31]. The most important mycotoxin from this group is ochratoxin A (OTA) [20]. OTA is very resistant to thermal treatment and stable in an acid environment.

OTAs are found in many foods such as maize seeds, wheat, barley, oats, rye, rice, flour, coffee, cocoa, legumes. They are found also in some fermented products, such as wine, beer, soy sauce. Dried fruits are also characterized by high OTA content. OTA can also contaminate animal products, such as milk and meat, especially pork, the consumption of which poses a risk of developing OTA nephropathy and urinary tract cancers. OTA tends to accumulate in animal tissues as it is fat soluble [32]. OTA is highly toxic, causes neurotoxic and teratogenic effects, impairs immunity, and primarily damages the kidneys. OTA interferes with the activity of phenylalanine hydroxylase in the kidneys and liver, resulting in the inhibition of normal protein synthesis. It also inhibits RNA and DNA synthesis [33]. The harmfulness of OTA is caused by irreversible damage to nephrons, which in turn may lead to death. In 1993, it was recognized by the International Agency for Research on Cancer (IARC) as possibly carcinogenic for humans [34,35].

Among farm animals, pigs are the most susceptible to the accumulation of OTA, whose tissue deposition occurs as follows: kidney > liver > muscle > fat. Even if OTA effects are known, the molecular mechanisms underlying the damage are still not completely clarified. OTA exposure (in vitro or in vivo) has been related to overproduction of reactive oxygen species (ROS), as well as oxidative damage (lipids, proteins, and DNA). In addition, OTA may reduce the antioxidant defense of cells by reducing GSH and cytoprotective enzymes [36].

OTA causes a dangerous disease called Balkan endemic nephropathy [37,38]. Moreover, for several years now, attention has been paid to the OTA contamination of white and red wines in Italy and other Mediterranean regions. The main species responsible for the production of this mycotoxin is the fungus *Aspergillus carbonarius* (Bainier) Thom, which grows on the fruit of grapes during harvest [39,40].

### 3.4. Trichothecenes

Trichothecenes are small, amphipathic molecules that can move passively across cell membranes. They are easily absorbed via the integumentary and gastrointestinal systems, allowing for a rapid effect on rapidly proliferating tissues [41]. Several dozen trichothecenes have been described; however, only DON, T-2, and HT-2 toxins have drawn attention for public health so far [42]. In terms of chemical structure, they are divided into four groups: A, B, C, and D. The type-A trichothecenes (T-2 and HT-2 toxins, diacetoxyscirpenol) have a functional group other than a ketone at carbon position 8. They occur less frequently but are more toxic than type-B trichothecenes [42]. The most common and well-studied is deoxynivalenol (DON), which usually occurs together with its two derivatives (3-acetyl and acetyl-deoxynivalenol). It mostly occurs as a result of infection with fungi of the genus *Fusarium* in maize, wheat, and other cereals grown in temperate climates [43]. Contact with DON in humans causes: vomiting, diarrhea, abdominal pain, headache, dizziness and fever [44,45]. It has been shown that infants and children are most susceptible to DON infestation [46].

Exposure to DON, as well as other trichothecenes, can have endocrine disruptive effects, such as reduced weight gain, neuroendocrine changes, and they also exert immune modulation. Acute exposure can cause leukocytosis, hemorrhage, and, with extremely high exposure, even death. This is the reason why, a few years ago, researchers raised questions about the safety of DON in exposed high-risk groups, including children and vegetarians [47].

### 3.5. Zearalenone

The sources of zearalenone (ZEN) are fungi of the genus *Fusarium*, often found in maize, wheat, barley, and other cereals [48]. Contamination of ZEN often occurs simultaneously with DON contamination and less frequently with AFs [42]. ZEN is stable at normal cooking temperatures and partially eliminated at high temperatures [49]. Due to its structural similarity to naturally occurring estrogens, it can be described as an estrogenic mycotoxin [20]. It causes changes in the reproductive system [48].

Because of its estrogenic properties, ZEN has been classified as a non-steroid estrogen or mycoestrogen, but its metabolites can show a 10-fold higher estrogenic activity than ZEN itself. Its major metabolites are zearalanone (ZAN), α-zearalenol (αZOL), β-zearalenol (βZOL), α-zearalanol (αZAL), and β-zearalanol (βZAL). Similar to trichothecenes, ZEN is rapidly absorbed from the gastrointestinal tract and metabolized. In animals, there are two main routes of ZEN biotransformation: hydroxylation, leading to the formation of αZOL and βZOL, and conjugation of ZEN with glucuronic acid. ZEN and its metabolites can cause changes in the metabolic profile of steroid-dependent cells, such as granulosa cells of ovarian follicles [50].

ZEN has a destructive effect on hormonal balance [51]. Depending on the dose, symptoms such as increased uterine weight and the number of antral follicles in an ovary have been observed in rats [52,53]. The estrogenic potency of both types of ZEA metabolites is different. α-ZOL demonstrates a higher binding capacity to estrogen receptors in comparison to the parent ZEA. In turn, β-ZOL has the lowest binding affinity of the three compounds. The species-specific rate of ZEA conversion to α-ZOL can be considered as a bio-activation reaction while ZEA transformation into β-ZOL as an inactivation reaction [54].

### 3.6. Fumonisins

Fumonisins (FB) are a family of more than 25 mycotoxins produced by fungi of the genus *Fusarium*, the most common of which are fumonisin B_1_ (FB_1_) and B_2_ (FB_2_) [55]. Maize seeds are the most susceptible to infection, although the presence of these mycotoxins can be also found in sorghum, wheat, barley, soybeans, asparagus, figs, and black tea. They are the most common mycotoxins in maize seeds [11]. FBs are hydrophilic mycotoxins. They are structurally different from most other mycotoxins that can be completely dissolved in organic solvents. Due to their hydrophilicity, FB does not get into milk, but small amounts accumulate in animal tissues, which are later used as raw materials for the food industry. FBs impair the functioning of the kidneys. They are also classified as neurotoxins because they damage the sphingosine biosynthetic pathway as a component of brain and nervous tissue [56]. FB_1_ can cause a variety of diseases in animals, for example equine leukoencephalomalacia, porcine pulmonary edema syndrome, hepatic tumor in rats, acute and fatal nephrotoxicity and hepatotoxicity in lambs. Varying degrees of toxic reactions (e.g., reduced weight gain, increased mortality, reduced size of the bursa of *Fabricius*, thymus, and spleen, myocardial degeneration, myocardial hemorrhage, alterations in the hemostatic mechanism and necrosis of hepatocytes) have been observed in chickens, ducks, and turkey chicks. Therefore, this mycotoxin not only pose a serious threat to human and animal health, but also affects food safety and limits animal production [57].

One of the paradoxes related to the toxicity of FB in animals concern the toxicokinetics of FB at the onset of mycotoxicosis. Most studies on animals concluded that FB toxicity is cumulative. In avian species, prolonged exposure of ducks and turkeys to low doses of FB resulted in a gradual increase in sphinganine (Sa) and sphingosine (So) bases over time in the liver; Sa and So being recognized markers of FB exposure and toxicity. However, toxicokinetic studies conducted both in avian species and in mammals revealed that FB is rapidly eliminated from the blood, and persistence of FB in animals was considered to be negligible. The apparent paradox between the cumulative toxicity of FB and their rapid plasma elimination may in fact be related to the lack of sensitivity of the analytical methods used [58].

### 3.7. Modified and Masked Mycotoxins

It has been noted for some time that the levels of mycotoxins found in food may be underestimated due to the presence of modified mycotoxins [59].

Modified mycotoxins are metabolites of mold fungi that normally are undetected when the parent mycotoxin is tested. These modified forms of mycotoxins can be produced by fungi or generated as part of an infected plant’s defense mechanism. In some cases, they are formed during food processing. The different processing steps influence the levels of mycotoxins present in the final product [60]. Research indicates that some modified mycotoxins can be converted into parent mycotoxins as a consequence of digestive processes in humans and animals, which can lead to adverse health effects. The resulting compounds may be even more toxic as long as their bioavailability is greater than that of the parent mycotoxin. Although the toxicological data are scarce, the possibility of converting a modified mycotoxin to its free form can pose a potential health risk, not only to humans but also to animals [2]. Masked mycotoxins are derivatives of mycotoxins that are undetectable by conventional analytical techniques due to the fact that their structures changed in the plant [60]. Conventional methods, for example ELISA, respond to masked forms, while this is unlikely with HPLC-based methods. It has been suggested that the analysis of the mycotoxin content in samples containing these compounds leads to their underestimation. Masked mycotoxins may not be detected during analyses due to altered physicochemical properties of their molecules [61]. Recognition that masked mycotoxins are of toxicological importance in food products suggests that generic toxicity estimates should be developed to be used by regulators and food producers to protect the health of consumers.

Of the masked mycotoxins, deoxynivalenol-3-glucoside (DON-3-G) and zearalenone sulfate (ZEA-S) are the most common in food and feed. Their toxicological properties are currently being investigated and the main concern is related to the conversion of DON-3-G to DON and ZEA-S to UAE by the microbiota of the gastrointestinal tract [62]. It is a group of chemically differentiated mycotoxins for which, to date, no legal regulations regarding safe levels in food and feed exist, and risk assessment studies are still ongoing. Moreover, there is no clear indication of the toxicity of other fungal secondary metabolites, which are often found in cereals, such as aurofusarin (AUR) and coulmorin (CULM), and are still under extensive research. In addition, there is a lack of information on the fate of masked, modified, and emerging mycotoxins and other secondary fungal metabolites in corn products and by-products [63], which are a significant raw material in the food industry, but are also raw material for the production of food and animal feed.

It is worth emphasizing that many structurally related substances, referred as modified mycotoxins, are generated by plant or fungal metabolism, as well as by food processing and coexist with their native forms [64]. As a result of their complex and variable chemical structure and ubiquitous presence, humans and animals may potentially be exposed to one or more mycotoxins or their forms modified by eating a contaminated diet. Studies [65] have shown that the presence of modified forms of mycotoxins is more often reported in food compared to feed. In addition to the presence of ZEN and its modified forms of phase I and phase II biotransformation, only a limited amount of quantitative data is available for the other modified forms, e.g., acetyl DON derivatives, hydrolyzed FBs, phase I metabolites T2 and NIV3G. Moreover, data are still scare and unevenly reported, despite increased awareness of the contribution of modified forms to the toxicity of mycotoxins. Overall, there have been recent promising advances in the field of analytical methods, which is a positive indicator of upcoming improvements in the simultaneous determination of multiple mycotoxins, both native and modified forms. However, analytical methods are still a limiting factor in complete data collection, both in terms of cost and lack of appropriate protocols [65].

## 4. Methods of Preventing Synthesis of Mycotoxins

The main strategy against the presence of mycotoxins in food, pet-food, and feed is to prevent the growth of the molds that produce them. Many methods are used today, from improved agricultural practices to traditional cultivation of resistant plant varieties and the use of genetic engineering. An important aspect turns out to be the monitoring of weather conditions conducive to the development of molds,;thanks to the maps of metrological data it is possible to predict the occurrence of the mycotoxin. Agrotechnical measures aimed at preventing the formation of mycotoxins include, inter alia, appropriate fertilization, and crop rotation. The soil is the main reservoir of spores for the fungi of the genus *Fusarium*. Their concentration in the substrate is increased by the continuous cultivation of one plant species, which results in an increased risk of plant infection [66]. A preventive measure is also the selection of plant varieties that are resistant to infection by mold fungi. Another example is the use of measures to protect plants against the growth of weeds and insects, because weeds provide a shelter for developing mold fungi, while insects are responsible for transferring fungal spores and damaging plant tissues and seeds, which facilitates the penetration of pathogens [67]. Timely harvesting with minimal moisture, and the proper storage of raw materials, are also important. In the European Union, agricultural practices are carried out in accordance with “Good Agricultural Practice” (GAP). Attention is paid to carrying out agrotechnical operations in such a way as not to cause the growth of the inoculum of fungi in agricultural crops and in production fields. In the case of already harvested raw materials, sorting them is the safest method of fighting mycotoxins. Optical sorting consists in separating damaged, discolored and generally abnormal-looking seeds, nuts, rice, legumes, fruits, etc., from those that show the correct characteristics for their species. The concentration of mycotoxins is significantly reduced as a result of washing and hulling the grains. Another method of eliminating mold fungi is the use of gamma and UV radiation, but this method does not reduce the level of mycotoxins that have already been produced, but only limits the development of fungi [68]. Chemical methods, in turn, rely on the use of chemical compounds to absorb, displace, or inactivate mycotoxins. Among others, ammonia, hydrogen peroxide, and sulfur (IV) oxide are used, although they are not of practical use because they change the nutritional value of the products and involve the risk of creating hazardous residues. Commercial additives (containing, for example, clays, zeolites, activated carbon) that can absorb toxins in the digestive tract are available, so that they are not absorbed into the bloodstream. In the fight against mycotoxins, biological methods are also used, consisting in the use of microorganisms characterized by the ability to remove various toxins from the environment. They work by metabolizing mycotoxins without the risk of producing side metabolites that are harmful to humans and animals. This method, however, is controversial among consumers, mainly due to the lack of legal regulations regarding this issue, however, studies clearly indicate its effectiveness [68].

High hopes are placed on the use of *Lactobacillus* and *Bifidobacterium* bacteria on a larger scale, which in addition to metabolizing mycotoxins have health-promoting properties, thanks to which they increase the nutritional value of products. *L. rhamnosus* strains were found to be the most effective in binding AF. Strains of *L. acidophilus* and *B. bifidum*, *B. longum* and others, reduce the amount of AFB_1_ [69]. It has been shown that not only AF is effectively cleared by lactic acid bacteria. The research also confirms the ability of bacteria belonging to the species *L. salivarius*, *L. delbrueckii* subsp. bulgaricus and *B. bifidum* to reduce the amount of OTA [70]. *Saccharomyces cerevisiae* yeast is characterized by similar properties, mainly OTA. These microorganisms are used on a large scale in many biotechnological processes, e.g., in baking, brewing, winemaking, and distilling. Due to the frequent mycotoxin contamination of the raw materials used in these processes (flour, malt, grape must), the possibility of fermentation using strains is considered, which, apart from the appropriate technological features, show the ability to reduce the toxin content, thus increasing the safety of the obtained product. Some yeasts also have probiotic features, which additionally resulting in an increased interest in the possibilities of their use [69].

The studies [71] showed that adding the Biobardin feed additive to the feed of broiler chickens in the amount of 5% may reduce the negative effects of the consumption of mycotoxins produced by the mold fungi *Fusarium graminearum, F. sporotrichiella, F. poae*, and *F. moniliformeobecne*. This addition in compound feed has an impact on the productivity of poultry and the prevention of mycotoxicity. Poultry suffering from chronic forms of mycotoxins were able to assimilate nutrients more efficiently than poultry that consumed similar feed free from the analyzed additives. It has been found that the consequences of chronic mycotoxicosis in poultry can be reduced by 25–50%.

Complete elimination of mycotoxins from feed is impossible, so effective mitigation programs are in place that can reduce the bioavailability of mycotoxins in the animal’s gastrointestinal tract. The use of natural products seems to be interesting. A study [72] evaluated the effect of various concentrations of the mixture of zeolite and bee brood in the diets of broilers under the influence of T-2 mycotoxin. It was shown that introducing a mixture of zeolite and bee brood into the diet of broiler chickens had a significant positive effect on the increase in live weight and blood parameters, reducing the negative effect of mycotoxins. Thus, the addition of zeolite and bee brood product to the diet has some protective effect when exposed to the T-2 mycotoxin produced by Fusarium fungi.

According to the World Health Organization (WHO), natural antioxidants can be a good choice for the prevention and treatment of various types of toxicity compared to other therapeutic agents in terms of low price, safety, and efficacy [73]. Dietary antioxidants are a potential source of neutralizing oxidative stress and maintaining redox homeostasis. Natural antioxidants can be vitamins (E and C), metalloproteins (ferritin, lactoferrin and albumin), other biologically active substances (flavonoids, carotenoids and anthocyanins), or minerals. One of the key approaches to protecting the body from oxidative damage is to incorporate natural antioxidants into the daily diet [73]. Therefore, dietary antioxidants advocate an innovative approach to protecting the body from a variety of toxins by strengthening the body’s internal antioxidant system against the negative effects of mycotoxins [74].

## 5. Food

The awareness of people around the world regarding safe and healthy food is growing, which is contributing to the creation of programs to detect and alert about the detection of hazardous substances. In the European Union, the food safety strategy includes the Rapid Alert System of Food and Feed (RASFF). The information obtained from the RASFF system enables risk prevention and early remedial action. The functioning of the system is based on the collection and quick dissemination of information about food products and feeds. Mycotoxin contamination of foodstuffs is the cause of a significant proportion of the notifications (around 20% in the European RASFF network) [75].

Due to their high consumption, cereals pose a particular threat to humans as a potential source of mycotoxins. Consumption of pseudocereals (quinoa and kañiwa) is currently increasing, although little is known about the susceptibility of these crops to mycotoxin contamination. The study by Ramos-Diaz et al. [76] determined the levels of mycotoxins and metabolites of mold fungi in Andean grains (quinoa and kañiwa) compared to cereal grains (barley, oats, and wheat), grown both in South America (Bolivia and Peru) and Northern Europe (Denmark, Finland, and Latvia). A total of 101 analytes at different levels were detected, mainly produced by fungi of the genera *Penicillium* spp., *Fusarium* spp. and *Aspergillus* spp. Their presence depended on the type of crop, geographic location, and agricultural practices used. In general, pseudocereals from South America were less contaminated with mycotoxins than those from Northern Europe, while the opposite was true for cereals. The mycotoxin contamination profiles showed significant differences between pseudocereals and cereals, even when harvested from the same regions.

Meat and meat products can also be contaminated with mycotoxins, especially animal organs, such as the kidneys and liver, which are the main organs implied in the detoxification of xenobiotics. Alaboudi et al. [77], analyzing frozen and fresh meat and offal (liver, kidneys), showed the presence of mycotoxins (AFB_1_, OTA, ZEN, FB_1_, and DON) in all samples. It should be emphasized that ZEN and AFB_1_ exceeded the MRLs of 14% and 2%, respectively, in the samples of frozen chicken muscles. The liver contained ZEN, and the kidneys contained ZEN, FB_1_, and DON. ZEN levels were higher in the liver (30.0%) than in the kidney samples (8.0%). Such analyses and data may be useful for state food control authorities to implement monitoring and control of mycotoxin residues in all poultry meat products. However, it is not a rule that all offal can be contaminated with mycotoxins, because, as shown by van Deventer et al. [78], a lack of mycotoxins was observed in marker tissues (liver and kidney) and in muscle tissue obtained from registered slaughterhouses in South Africa.

Due to the sensitivity of infants, safe food is extremely important, especially in their case. Mammalian milk may contain contaminants from maternal exposure. Memis et al. [79] showed the presence of AFM_1_, OTA, ZEN, and DON in human milk. It was also shown that higher levels of OTA were associated with exposure to smoking (environmental, maternal smoking). The conclusion is that mycotoxins can pass into breast milk and maternal exposure to smoking may have an impact on this situation. Exposure to mycotoxins mentioned above can also lead to lactation problems.

The study of milk substitutes [80] found the presence of 17 mycotoxins, including AFB_1_, ZEN, DON, and FB_1_. Infant formulas were less contaminated than grain products. It was found that the limit for baby food for AFB_1_ was exceeded in flour. Interestingly, two toxins not previously described in the literature, namely aflatoxicol and sterigmatocystin, were identified in 3% and 17% of baby food, respectively.

## 6. Pet Food and Feed

Apart from humans, animals can also be exposed to microbiological hazards. This is especially possible for companion animals, especially dogs and cats. Even dried dog snacks can be a source of such contaminants. Pet food may also be a hazard, and not only because of bacterial contamination [81]. Currently, research is being conducted to detect the contamination of mycotoxins in pet food. Pigłowski’s analyses showed [75] that these substances were the most frequently reported hazard category in the RASFF system from 1981 to 2017. Importantly, the majority of reports concerned AFB_1_ [75]. In the case of animal food, the mycotoxins of most concern are AFs, OTA, ZEN, and FB [82]. However, a large group of EU recommendations for safe levels in animal products only apply to three mycotoxins (Table 2) [83].

It has been shown that, in the case of cereals, most of the impurities are close to the grain surface. This means that removing only part of the outer layers of the grains can reduce mycotoxin contamination [84]. Research [85] showed that metabolites of lactic acid bacteria (LAB) can be a valuable alternative in reducing fungal infections before and after harvest. It has been shown that mycotoxins present in food can lead to a reduction in food consumption and have a negative impact on animal health [86,87,88]. The presence of these substances may also inhibit overall weight gain, for example, in domestic animals [89].

In addition, the presence of mycotoxins in edible animal products, such as milk, meat, and eggs, is possible, which may have long-term negative health effects on humans [90,91]. Contamination with fungi and mycotoxins affects both the organoleptic properties and the nutritional value of the pet food and carries the risk of poisoning. Moreover, studies have shown that a high percentage of pet food samples are contaminated with more than one mycotoxin [92].

The individual sensitivity of animals, the amount of toxins present and the time of exposure are the main factors determining the effects of consuming food contaminated with mycotoxins [92,93]. AFB_1_ has been shown to have a strong hepatotoxic effect, but it can be minimized, for example, by supplementing the food with curcumin [94]. Control of the production of mycotoxins and the growth of molds responsible for their formation should be a priority issue for food producers, both for humans and animals [95]. A study by Singh and Chuturgoon [96], the purpose of which was to compare the microbiological quality of supermarket pet foods with premium foods, showed that regardless of the manufacturer, all food samples were contaminated with fungi (mainly *Aspergillus flavus*, *Aspergillus fumigatus* and *Aspergillus parasiticus*), as well as their secondary metabolites (most often AFs and FBs). The results of these studies indicate that more-expensive dog food does not provide the highest quality, and does not guarantee microbiological purity. It has been shown that in the case of dry animal food, the extrusion process can reduce the pathogenicity of microorganisms without affecting the digestibility of the food [97]. In subsequent studies focusing on the microbiological assessment of foods, a disturbing presence of FBs was found [98].

Mycotoxins are substances that are difficult to monitor continuously and a universal risk assessment tool would help to assess if there is a particular risk due to the inclusion of certain feed ingredients. To this end, a study [99] estimated the toxin content of 97 commercial fish feeds, with the most significant toxins in fish feed being DON, ZEN, FB and enniatin. They pose a risk to the welfare of fish, which can be calculated using Bayesian models to determine the 5% critical concentrations (CC5) for various toxins. Bayesian network (BN) modeling is one of the widespread machine learning modeling techniques, and can deal very well with both unbalanced data and missing data. BN models are developed on the basis of observational data. Such models make predictions by computing conditional probabilities among the available variables in the dataset [100].

In addition to fish meal, wheat, soy products, and corn are regularly used as fish feed ingredients. The calculated scenarios show that fish are at high risk of toxin contamination if low-quality feed ingredients are selected for feed production. It is therefore necessary to set specific maximum levels for several mycotoxins in fish feed [99].

Twarużek et al. [101] assessed the level of mycotoxin contamination of raw materials and products for animals in Poland in 2015–2020. Producers, farmers and veterinarians provided a total of 3980 samples (642 maize samples, 2027 feed samples, 990 fine grain samples, 142 maize silage samples and 179 TMR samples). The samples were analyzed for the presence of several mycotoxins, including AFs, FB, OTA, DON, ZEN, T-2 toxin, and H-2 toxin. Studies have shown that DON and ZEN were the most common contaminants in maize samples (97.3% and 98.4%, respectively) and feed (99.7% and 100% samples, respectively). They were also present in all maize and TMR silage samples. The highest concentrations of DON and ZEN were 16,889 μg/kg in the wheat sample and 1420 μg/kg in the maize sample. Additionally, in 51 trials, the level of mycotoxins (mainly DON and ZEN) exceeded those recommended by the European Union. The present study showed that both feed and raw materials are contaminated with mycotoxins, often by more than one [101].

Contamination of feeds with several types of FBs at the same time is quite common. It should be borne in mind, that mycotoxins show a synergistic effect. Especially the coexistence of FB with other toxins produced by *Fusarium* sp. Increases the risk. The research by Witaszak et al. [89] confirmed the presence of five mycotoxins in the amounts allowed by EU regulations. Despite this, caution should be paid because low levels of mycotoxins do not eliminate the risk in dog food, and long-term daily consumption of even small amounts of mycotoxins can lead to slow damage to the animal’s body and the development of many diseases, including cancer. Mycotoxins were also found in food in the studies of Shao et al. [102]. Only one out of 32 samples was free of mycotoxin contamination. Moreover, all other samples were contaminated with at least three different types of mold secondary metabolites. Research by Tegzes et al. [103] aimed to compare grain and grain-free dog food in terms of mycotoxin content. The results of these analyzes confirmed the presence of mycotoxins in dry grain foods for dogs, while they were not found in grain-free foods. These analyses suggest that the risk of exposure to mycotoxins is higher with grain-based dry dog food. To minimize risk, dog food manufacturers should select grain types that are less susceptible to mycotoxins [104].

The main ingredient of veterinary therapeutic diets (VTD) is grain. These foods are intended for dogs with various diseases that require safe and nutritious nutrition. However, it is disturbing that cereal grains can quite often be contaminated with *Fusarium* fungi, which are responsible for the production of mycotoxins. In the studies by Witaszak et al. [105], samples of VTD were analyzed for the presence of molds and their secondary metabolites. Among the analyzed samples, only 9.5% were free from mycotoxins produced by *Fusarium*, however, none of the tested samples exceeded the permissible limits of mycotoxins content in pet food, as defined by EU regulations. It should be kept in mind, that systematic testing of both domestic and VTD for the content of harmful microorganisms and their metabolites is necessary, because especially VTD should be characterized by the highest level of safety for animals [81].

The research of Macias-Montes et al. [106] shows that the presence of mycotoxins is fairly common in dry dog food, however, the concentrations of most of them were among the lowest recorded so far. These studies showed that the food quality had no influence on the mycotoxin content. The problem may be chronic exposure to mycotoxins and their modified forms. The results of the research by Okuma et al. [107] revealed a low prevalence of AF and OTA in commercial pet foods. Although DON has been detected in many trials, its levels were well below those likely to cause acute toxic effects. On the other hand, studies by Gazzotti et al. [108] showed that all samples of extruded complete dog food complied with current European legislation on mycotoxin limits. However, these results reiterated the need for further research into the potential risk of chronic low-dose exposure to various types of mycotoxins to which pet species are currently exposed [81].

An important risk is seizure-forming mycotoxins. Penitrem A (PA) and roquefortine (RQ) are synthesized by some molds (*Penicillium* spp., *P. crustosum* and *P. roqueforti*). Their source is moldy food, mainly cereals and their products, e.g., bread, and various types of nuts (walnuts, almonds, peanuts), possibly also moldy dairy products. The cause of dog poisoning with roquefortine may also be ripening cheese with blue mold. This toxin is rapidly absorbed from the gastrointestinal tract. Its neurotoxic effect appears shortly after exposure [109,110]. Therefore, in connection with the above information, it is worth paying attention to the content of home garbage cans and garden composters. Perishable food that produces fungal toxins can pose a serious risk to companion animals.

In recent years, there has been a growing tendency to eat organic food instead of conventional food. This trend is mainly due to concerns about the potential negative health effects of consuming products containing pesticide residues, fertilizers, hormones and antibiotics, which are widely used in regular food production. While the legal acts governing the cultivation of organic raw materials prohibit the use of these products, environmental pollution can occur in both conventional and organic foodstuffs [111]. In the case of animal nutrition, grain-free food may turn out to be a safer solution in terms of mycotoxin content [103], although among pet food, especially the “premium” ones, raw materials from organic farming are more and more often used, and food from agriculture organic is considered safer in terms of pesticide content, but avoiding insecticides and fungicides in organic farming can lead to the growth of fungi and the formation of mycotoxins. This increases the likelihood of their occurrence in organic products, while increasing the consumer’s exposure to these risks. Therefore, the safety function that has been assigned to organic food worldwide may be questionable depending on the potential environmental contamination of that food [112].

## 7. Conclusions

Both humans and animals are daily at risk of mycotoxins [5,6,7]. Due to the large amount of grain products consumed, it becomes extremely important to check them regularly for safety and to detect harmful substances. One of the most important indicators of the quality of food and pet food and feed is the content of mycotoxins in them. They are a threat that we are not able to completely eliminate from the surrounding environment. Their occurrence is favored by factors commonly present in our environment, including humidity. Modified mycotoxins may be a particularly significant risk. However, no legal regulations regarding safe levels in food and feed exist, and risk assessment studies are still ongoing. Mycotoxins show resistance to technological processes and have the ability to accumulate in tissues. They lead to economic losses and, in addition, they cause a wide variety of diseases for people and animals. Sometimes they can be the cause of death. As far as mycotoxins are concerned, milk, eggs, and meat do not constitute the main threat to human and animal health and life from the literature data. In most cases, their main source for humans and animals are eating contaminated grains and legumes and their products, and to a lesser extent food of animal origin. Research has shown that food products (such as cereal products, dried fruits, herbs and spices, wine) available on the market meet the applicable requirements for mycotoxin contamination and do not pose a threat to the health of consumers. However, the problem is mainly noticed in tropical and developing countries. The main strategy against the presence of mycotoxins in food, pet-food and feed is to prevent the growth of the molds that produce them. Many methods are used today, from improved agricultural practices to traditional cultivation of resistant plant varieties and the use of genetic engineering. Innovative methods include the use of zeolite, antioxidants and bacteria. Despite the fact that research on the content of mycotoxins in products is capital-intensive, it seems necessary to conduct monitoring in this area. In order to minimize the risk to the health of pets, the priority should be prevention, i.e., systematic testing of the raw materials and foods, in terms of the content of harmful microorganisms and their metabolites.

## Figures and Tables

**Figure 1 toxins-13-00822-f001:**
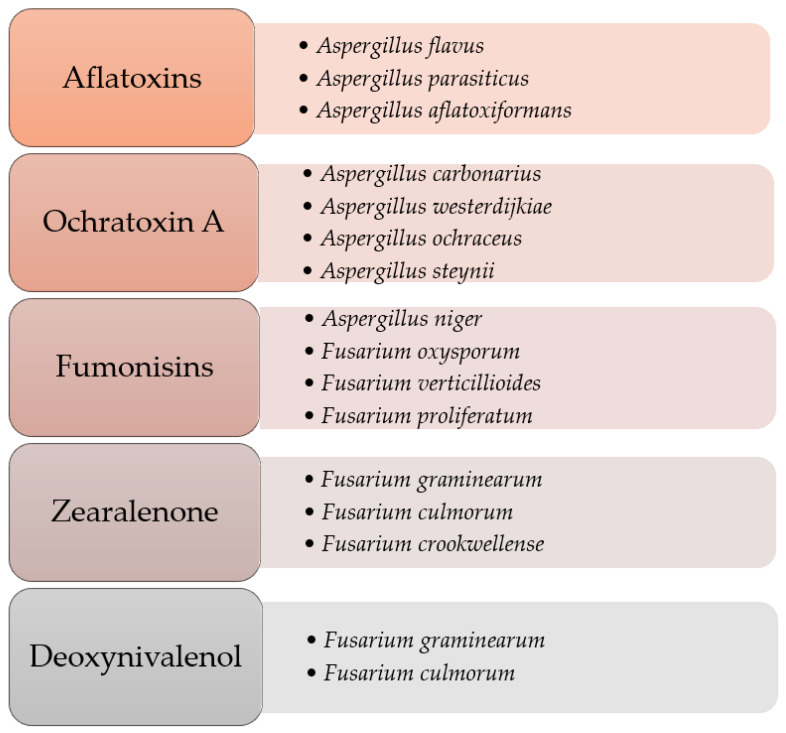
Selected species of mold fungi producing mycotoxins. Adapted from [10,11,12].

**Table 1 toxins-13-00822-t001:** Maximum levels for certain mycotoxins in selected foods. Adapted from [18].

Mycotoxins	Food Products/Raw Material	Maximum Level [μg/kg]
Sum of aflatoxin B_1_, B_2_, G_1_ and G_2_	Groundnuts and nuts and processed products thereof, intended for direct human consumption or use as an ingredient in foodstuffs	4
	Maize to be subjected to sorting or other physical treatment before human consumption or use as an ingredient in foodstuffs	10
	Dried fruit and processed products thereof, intended for direct human consumption or use as an ingredient in foodstuffs	4
Ochratoxin A	Unprocessed cereals	5
	Roasted coffee beans and ground roasted coffee, excluding soluble coffee	5
	Processed cereal-based foods and baby foods for infants and young children	0.5
Deoxynivalenol	Unprocessed cereals	1250–1750
	Pasta (dry)	750
	Bread (including small bakery wares), pastries, biscuits, cereal snacks and breakfast cereals	500
	Processed cereal-based foods and baby foods for infants and young children	200
Zearalenone	Bread (including small bakery wares), pastries, biscuits, cereal snacks and breakfast cereals, excluding maize snacks and maize based breakfast cereals	50
	Processed maize-based foods for infants and young children	20
Sum of fumonisins B_1_ and B_2_	Unprocessed maize	2000
	Processed maize-based foods and baby foods for infants and young children	200

**Table 2 toxins-13-00822-t002:** Guideline limit values for deoxynivalenol, zearalenone, and ochratoxin A in pet products. Adapted from [83].

Mycotoxin	Pet Food Product	Guide Value in mg/kg for a Pet Product with a Moisture Content of 12%
Deoxynivalenol	cereals and cereal products with the exception of maize by-products	8
	maize-by products	12
	compound feed	5
Zearalenone	cereals and cereal products with the exception of maize by-products	2
	maize-by products	3
	compound feed for adult dogs and cats other than those intended for reproduction	0.2
	compound feed for puppies, kittens, dogs and cats intended for reproduction	0.1
Ochratoxin A	cereals and cereal products	0.25
	compound feed for dogs and cats	0.01

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
