# Peer review of "Alimentary Risk of Mycotoxins for Humans and Animals"

_toxins, 2021, doi:10.3390/toxins13110822_

Round 1

Reviewer 1 Report

In the review “Nutritional risk of mycotoxins for human and animals” the authors overview the risk related to mycotoxins contaminations in food and feed. The review has been improved with respect to the previous version. Moreover, some issues should be still raised. The most relevant of them is that an imbalance in the description of the impact of mycotoxins on food and feed with respect to pet-food is evident. For this reason, I suggest the authors: a) limit the review to the pet-food (a very well described and documented section!) or b) implement the section devoted to food and feed.

General considerations

I suggest moving the lines 78-124 as a separate section whit the title “Food” before the chapter “Pet food and feed”. The description of the RASFF (lines 405-417) can be moved at the beginning of the Food section and summarized (see comments below).

I suggest considering the description of mycotoxins (from line 141) as a separate chapter (Chapter 3) and adding a brief introduction.

Please check all the acronyms in order to be sure to have spelt them out before in the text, and if an acronym has already been used before using it instead of the complete word (e.g. the first time you write the term aflatoxins you should put AF in brackets (AF), and then use AF throughout the text).

Other comments

Title: change “Nutritional” in “Alimentary”

Abstract

Line 9: change “may” in “can”

Line 11-12: remove the sentence “to….toxins”

Introduction

Line 25: remove “their” before “metabolites”

Line 26: remove “products”

Line 26-27: change the sentence “altered….biological” in “altered chemical structures with different physicochemical, chemical and biological properties”.

Line 31: add “such as” before “high stability”

Line 31: add “their” before “high toxicity”

Line 38: remove “level”

Line 39: add “can be detected” after “growth of animals”

Line 42: remove “the” before “diseases”

Line 45: change “study” in “review”, and “characterize” in “highlight”

Line 49: remove “in food” from the title

Line 56: remove “air” before “humidity”; Which are the “appropriate temperature”?

Line 67: add “characterized by” after “are”; change “can get into the body” in “can be absorbed”

Line 83: to what 33% is referred to?

Line 84: change “consumption” in “ingestion”

Line 86: remove “their” before “high”

Line 89: change “the study [20]” by citing the author

Line 100: change “which the body” in “which are the main organs implied in the detoxification of xenobiotics”.

Line 107: It is “state” for “country”?

Line 109: change “will” in “can”

Line 110: change “the lack” in “a lack”

Line 119: the authors state that “Exposure to mycotoxins can 118 also lead to lactation problems”. Which one?

Line 127: I suggest considering this chapter as part of chapter 2 (2.1)

Line 129: change “are” in “can be”

Line 134: change “leading to damage” in “affecting”

Line 137; What do the authors mean with “were separated”?

Line 142: change the sentence “aflatoxins…..distinguished” in “Different classes of aflatoxins (AF), such as B1, B2, G1 and G2 can be recognised”. Change “markings” in “capital letters”

Line 144: add “compound” after “toxic”

Line 158: remove the sentence “that the body has come into contact with”

Line 165: change “kept” as “classified” and “under” as “in”

Line 167: change “at” in “in”

Line 184-185: remove the sentence “AFM1…..AFB1”

Line 194: change “;” in “:”

Line 195: change “legume seeds” in “legumes”

Line 220: change “inhabits” in “growth on”

Line 225: remove “of ingested thricothecenes”

Line 240: change “infection” in “infestation”

Line 242: add “also exert” before “immune modulation”

Line 254: change the sentence “when exposed to high-risk groups” in “in exposed high-risk groups”

Line 254: What do the authors mean with “much like than naturally-occurring estrogens”?

Line 257: change “derivatives” in “metabolites”

Line 260: change “alimentary canal” in “gastrointestinal tract”

Line 266: what do the authors mean by “vaginal opening” and “enlargement of uterus at periphery”?

Line 269-270: Why do the authors affirm that “The 269 precise mechanism of the reproductive toxicity of ZEA has not been established yet”? It is well recognised that ZEA acts by binding to estrogen receptors. I think that some more explanation about the mechanism of action should be added.

Line 290: remove the second “increased mortality”

Line 318: change “sparse” in “scarce”

Line 339: add “and” before “crop rotation”

Lines 362-363: change the sentence “Commercial additives (containing, for example, clays, zeolites, activated carbon) are available that can absorb toxins in the digestive tract” in “Commercial additives (containing, for example, clays, zeolites, activated carbon) that can absorb toxins in the digestive tract are available”

Line 364: remove “from the food”

Line 405-418: change as “The awareness of people around the world about safe and healthy food is growing, which is contributing to the creation of programs to detect and alert about the detection of hazardous substances. In the European Union, the food safety strategy includes the Rapid Alert System of Food and Feed (RASFF). The information obtained from the RASFF system enables risk prevention and early remedial action. The functioning of the system is based on the collection and quick dissemination of information about food products and feeds. Mycotoxin contamination of foodstuffs is the cause of a significant proportion of the notifications (around 20% in the European RASFF network”

Line 219: remove “An important fact is that”

Lines 439-441 seem to be referred to food-producing animals and not pet as well as lines 468-490. I suggest moving them, because starting from line 492 the authors start again to write about pet-food.

Line 496: change “exercised” in “paid”

Author Response

Dear Reviewer,

We would like to kindly thank you for the insightful review of our manuscript. Thank you for the constructive comments and suggestions. Below we attached the list of changes made according to your suggestions. In the revised version of the manuscript we have marked the corrected parts of the text in the track change mode.

Yours sincerely,

Reviewer:

Comments to the Author

In the review “Nutritional risk of mycotoxins for human and animals” the authors overview the risk related to mycotoxins contaminations in food and feed. The review has been improved with respect to the previous version. Moreover, some issues should be still raised. The most relevant of them is that an imbalance in the description of the impact of mycotoxins on food and feed with respect to pet-food is evident. For this reason, I suggest the authors: a) limit the review to the pet-food (a very well described and documented section!) or b) implement the section devoted to food and feed.

Point 1: I suggest moving the lines 78-124 as a separate section whit the title “Food” before the chapter “Pet food and feed”. The description of the RASFF (lines 405-417) can be moved at the beginning of the Food section and summarized (see comments below).

Response: We moved these lines and separated as chapter “Food” as suggested. We also moved the description of the RASFF.

Point 2: I suggest considering the description of mycotoxins (from line 141) as a separate chapter (Chapter 3) and adding a brief introduction.

Response: We left it unchanged, because in chapter 3. Harmful effect of mycotoxins for humans and animals, after a short subsection 3.1. Classification, further from 3.2 to 3.7 is the division into individual groups of mycotoxins.

Point 3: Please check all the acronyms in order to be sure to have spelt them out before in the text, and if an acronym has already been used before using it instead of the complete word (e.g. the first time you write the term aflatoxins you should put AF in brackets (AF), and then use AF throughout the text).

Response: We corrected it as suggested.

Point 4:

Other comments

Title: change “Nutritional” in “Alimentary”; Line 9: change “may” in “can”; Line 11-12: remove the sentence “to….toxins”; Line 25: remove “their” before “metabolites”; Line 26: remove “products”; Line 26-27: change the sentence “altered….biological” in “altered chemical structures with different physicochemical, chemical and biological properties”.; Line 31: add “such as” before “high stability”; Line 31: add “their” before “high toxicity”; Line 38: remove “level”; Line 39: add “can be detected” after “growth of animals”; Line 42: remove “the” before “diseases”; Line 45: change “study” in “review”, and “characterize” in “highlight”; Line 49: remove “in food” from the title

Response: Thank you for your valuable comment, we made these changes.

Point 5: Line 56: remove “air” before “humidity”; Which are the “appropriate temperature”?

Response: We changed “appropriate” to “optimal”

Point 6: Line 67: add “characterized by” after “are”; change “can get into the body” in “can be absorbed”

Response: Thank you for your valuable comment, we made these changes.

Point 7: Line 83: to what 33% is referred to?

Response: We deleted it

Point 8: Line 84: change “consumption” in “ingestion”; Line 86: remove “their” before “high”; Line 89: change “the study [20]” by citing the author; Line 100: change “which the body” in “which are the main organs implied in the detoxification of xenobiotics”.; Line 107: It is “state” for “country”; Line 109: change “will” in “can”; Line 110: change “the lack” in “a lack”

Response: Thank you for your valuable comment, we made these changes.

Point 9: Line 119: the authors state that “Exposure to mycotoxins can 118 also lead to lactation problems”. Which one?

Response: We added information – “Exposure to mentioned above mycotoxins can also lead to lactation problems.” (line 438)

Point 10: Line 127: I suggest considering this chapter as part of chapter 2 (2.1)

Response: We did it.

Point 11: Line 129: change “are” in “can be”; Line 134: change “leading to damage” in “affecting”

Response: Thank you for your valuable comment, we made these changes.

Point 12: Line 137; What do the authors mean with “were separated”?

Response: We changed this sentence.

Point 13: Line 142: change the sentence “aflatoxins…..distinguished” in “Different classes of aflatoxins (AF), such as B1, B2, G1 and G2 can be recognised”. Change “markings” in “capital letters”; Line 144: add “compound” after “toxic”; Line 158: remove the sentence “that the body has come into contact with”; Line 165: change “kept” as “classified” and “under” as “in”; Line 167: change “at” in “in”; Line 184-185: remove the sentence “AFM1…..AFB1”; Line 194: change “;” in “:”; Line 195: change “legume seeds” in “legumes”; Line 220: change “inhabits” in “growth on”; Line 225: remove “of ingested thricothecenes”; Line 240: change “infection” in “infestation”; Line 242: add “also exert” before “immune modulation”; Line 254: change the sentence “when exposed to high-risk groups” in “in exposed high-risk groups”

Response: Thank you for your valuable comments, we made these changes.

Point 14: Line 254: What do the authors mean with “much like than naturally-occurring estrogens”?

Response: We decided to delete it.

Point 15: Line 257: change “derivatives” in “metabolites”; Line 260: change “alimentary canal” in “gastrointestinal tract”

Response: Thank you for your valuable comment, we made these changes.

Point 16: Line 266: what do the authors mean by “vaginal opening” and “enlargement of uterus at periphery”?

Response: We deleted this sentence.

Point 17: Line 269-270: Why do the authors affirm that “The 269 precise mechanism of the reproductive toxicity of ZEA has not been established yet”? It is well recognised that ZEA acts by binding to estrogen receptors. I think that some more explanation about the mechanism of action should be added.

Response: We added some more information about the action of ZEN (lines 220-225).

Point 18: Line 290: remove the second “increased mortality”; Line 318: change “sparse” in “scarce”; Line 339: add “and” before “crop rotation”; Lines 362-363: change the sentence “Commercial additives (containing, for example, clays, zeolites, activated carbon) are available that can absorb toxins in the digestive tract” in “Commercial additives (containing, for example, clays, zeolites, activated carbon) that can absorb toxins in the digestive tract are available”; Line 364: remove “from the food”; Line 405-418: change as “The awareness of people around the world about safe and healthy food is growing, which is contributing to the creation of programs to detect and alert about the detection of hazardous substances. In the European Union, the food safety strategy includes the Rapid Alert System of Food and Feed (RASFF). The information obtained from the RASFF system enables risk prevention and early remedial action. The functioning of the system is based on the collection and quick dissemination of information about food products and feeds. Mycotoxin contamination of foodstuffs is the cause of a significant proportion of the notifications (around 20% in the European RASFF network”; Line 219: remove “An important fact is that”

Response: Thank you for your valuable comments, we made these changes.

Point 19: Lines 439-441 seem to be referred to food-producing animals and not pet as well as lines 468-490. I suggest moving them, because starting from line 492 the authors start again to write about pet-food.

Response: These lines refer to raw materials for pet-food for e.g. dogs and cats, that is why it is there.

Point 20: Line 496: change “exercised” in “paid”

Response: Thank you for your valuable comment, we made these changes.

Reviewer 2 Report

The paper mainly expounds the health risks caused by mycotoxins to humans and animals, and introduces the composition, toxicity and occurrence of important mycotoxins. The structure of the article is relatively complete and clearly organized. The literature analysis method is used to summarize the research, and the research work is presented concisely and clearly in the form of diagrams. The content of the review provides a certain reference value for scholars to understand the health risks of mycotoxins, and to improve people's awareness of safe and healthy food. I think this work could be accepted after some minor revisions.

Following are some comments

  1. The research work of the thesis needs to add some research content. For example:
    • For 136, 179 and 211, they are reviewed from the classification, toxicity and carcinogenic mechanism, and cited literature examples. In the discussion of 232, 255, there is a lack of literature examples to support.
    • Paper 310 focuses on methods to prevent the synthesis of fungal toxins, including improved agricultural practices, the selection of drug-resistant plant varieties and physical, chemical and biological methods, but lacks detailed elaboration, especially on the main subjects of the paper (food, pet food and feed).
    • Article 286's introduction to masking mycotoxins is a bright spot. The author needs to provide more specific research literature and make a review and summary.
  2. It is suggested to adjust the layout of the article, adjust the content arrangement of 310 and 368, increase the literature discussion of specific research subjects (food, pet food and feed), enhance the relevance, and make the discussion more rigorous and powerful.
  3. The paper's focus of 427 relates to the significance of this work in the area of mycotoxin monitoring risk assessment. I think the author did not clearly explain the 431 Bayesian model as a risk assessment tool. The author quoted a related document, but I still don't know what happened.
  4. It is suggested that conclusion 515 should be deleted and supplemented. The conclusion is not only a summary of the key results of the study, but also a lack of systematic research recommendations on reducing mycotoxins in specific research objects (food and pet food and feed) that threaten human and animal health. The article does not highlight the insight and applicability of your findings for further work.
  5. Consider eliminating multiple references, such as section 31,205,398, and suggesting that the piles of literature be eliminated by describing each reference separately. Each reference can be summarized into 1-2 phrases to illustrate.
  6. Both figures and tables must come immediatelyafter the first time it is mentioned in the text, not far away. There is a paragraph gap between 55 picture abstracts and 58 in the article.

Author Response

Dear Reviewer,

We would like to kindly thank you for the insightful review of our manuscript. Thank you for the constructive comments and suggestions. Below we attached the list of changes made according to your suggestions. In the revised version of the manuscript we have marked the corrected parts of the text in the track change mode.

Yours sincerely,

Reviewer:

Comments to the Author

The paper mainly expounds the health risks caused by mycotoxins to humans and animals, and introduces the composition, toxicity and occurrence of important mycotoxins. The structure of the article is relatively complete and clearly organized. The literature analysis method is used to summarize the research, and the research work is presented concisely and clearly in the form of diagrams. The content of the review provides a certain reference value for scholars to understand the health risks of mycotoxins, and to improve people's awareness of safe and healthy food. I think this work could be accepted after some minor revisions.

Following are some comments

Point 1: The research work of the thesis needs to add some research content. For example:

For 136, 179 and 211, they are reviewed from the classification, toxicity and carcinogenic mechanism, and cited literature examples. In the discussion of 232, 255, there is a lack of literature examples to support.

Paper 310 focuses on methods to prevent the synthesis of fungal toxins, including improved agricultural practices, the selection of drug-resistant plant varieties and physical, chemical and biological methods, but lacks detailed elaboration, especially on the main subjects of the paper (food, pet food and feed).

Article 286's introduction to masking mycotoxins is a bright spot. The author needs to provide more specific research literature and make a review and summary.

Response: We added appropriate references to these lines (e.g. 48, 54, 55). We also added details about masking and modified mycotoxins (lines 281-309).

Point 2: It is suggested to adjust the layout of the article, adjust the content arrangement of 310 and 368, increase the literature discussion of specific research subjects (food, pet food and feed), enhance the relevance, and make the discussion more rigorous and powerful.

Response: Thank you for your valuable comment, we changed the layout of the article.

Point 3: The paper's focus of 427 relates to the significance of this work in the area of mycotoxin monitoring risk assessment. I think the author did not clearly explain the 431 Bayesian model as a risk assessment tool. The author quoted a related document, but I still don't know what happened.

Round 2

Reviewer 1 Report

I have truly appreciated the substantial improvement of the Review by the authors. Some minor issues should be raised (see below) but in my opinion, the Review can be accepted for publication under very minor concerns.

line 10: the word "animals" should be left in the text

line 23: remove the "-"

Line 24: change "with" in "and"

Line 92: remove "to" before "the central.."

Line 147: remove "and pig"

line 201: change as "...simultaneously with DON contamination and less frequently..."

Line 233: change "-its small" in "but small"

Line 281: change as "and main concern is related to conversion..."

line 284: remove "ongoing" before "risk assessment"

Line 291: remove "to" before "as"

Line 298: add "biotransformation" after "phase II"

Line 300: change "sparse" in "scarce"

Line 305: remove "the" before "lack"

Line 369: remove "in them".

Line 375: change "The study" in "A study"

Lines 419-420, lines 464-465, lines 504-505 and lines 520-521: use acronyms for mycotoxins

Line 602 remove "ongoing" before "risk assessment"

Author Response

Dear Reviewer,

We would like to kindly thank you for the insightful review of our manuscript. Thank you for the constructive comments and suggestions. Below we attached the list of changes made according to your suggestions. In the revised version of the manuscript we have marked the corrected parts of the text in the track change mode.

Yours sincerely,

Jagoda Kępińska-Pacelik and co-author

Reviewer:

Comments to the Author

I have truly appreciated the substantial improvement of the Review by the authors. Some minor issues should be raised (see below) but in my opinion, the Review can be accepted for publication under very minor concerns.

Point 1: line 10: the word "animals" should be left in the text

Response: We left word “animals”

Point 2: line 23: remove the "-"

Response: We remowed the “-“

Point 3: Line 24: change "with" in "and"

Response: We changed the word.

Point 4: Line 92: remove "to" before "the central.."

Response: We removed the word “to”

Point 5: Line 147: remove "and pig"

Response: We removed “and pig”

Point 6: line 201: change as "...simultaneously with DON contamination and less frequently..."

Response: We made changes

Point 7: Line 233: change "-its small" in "but small"

Response: We made these changes

Point 8: Line 281: change as "and main concern is related to conversion..."

Response: We made this change

Point 9: line 284: remove "ongoing" before "risk assessment"

Response: We removed it

Point 10: Line 291: remove "to" before "as"

Response: We removed it

Point 11: Line 298: add "biotransformation" after "phase II"

Response: We did it

Point 12: Line 300: change "sparse" in "scarce"

Response: We change the word

Point 13: Line 305: remove "the" before "lack"

Response: We removed “the”

Point 14: Line 369: remove "in them".

Response: We removed “in them”

Point 15: Line 375: change "The study" in "A study"

Response: We made this change

Point 16: Lines 419-420, lines 464-465, lines 504-505 and lines 520-521: use acronyms for mycotoxins

Response: We added acronyms in these lines

Point 17: Line 602 remove "ongoing" before "risk assessment"

Response: We removed it

We kindly thank you for these valuable comments and suggestions. Thanks to these comments, we improved our manuscript.